# Structural and Functional Peculiarities of α-Crystallin

**DOI:** 10.3390/biology9040085

**Published:** 2020-04-23

**Authors:** Olga M. Selivanova, Oxana V. Galzitskaya

**Affiliations:** 1Institute of Protein Research, Russian Academy of Sciences, 142290 Pushchino, Moscow Region, Russia; seliv@vega.protres.ru; 2Institute of Theoretical and Experimental Biophysics, Russian Academy of Sciences, 142290 Pushchino, Moscow Region, Russia

**Keywords:** α-Crystallin, chaperone, aggregation, adaptation, cataract, gel-like, amyloids

## Abstract

α-Crystallin is the major protein of the eye lens and a member of the family of small heat-shock proteins. Its concentration in the human eye lens is extremely high (about 450 mg/mL). Three-dimensional structure of native α-crystallin is unknown. First of all, this is the result of the highly heterogeneous nature of α-crystallin, which hampers obtaining it in a crystalline form. The modeling based on the electron microscopy (EM) analysis of α-crystallin preparations shows that the main population of the α-crystallin polydisperse complex is represented by oligomeric particles of rounded, slightly ellipsoidal shape with the diameter of about 13.5 nm. These complexes have molecular mass of about 700 kDa. In our opinion, the heterogeneity of the α-crystallin complex makes it impossible to obtain a reliable 3D model. In the literature, there is evidence of an enhanced chaperone function of α-crystallin during its dissociation into smaller components. This may indirectly indicate that the formation of heterogeneous complexes is probably necessary to preserve α-crystallin in a state inactive before stressful conditions. Then, not only the heterogeneity of the α-crystallin complex is an evolutionary adaptation that protects α-crystallin from crystallization but also the enhancement of the function of α-crystallin during its dissociation is also an evolutionary acquisition. An analysis of the literature on the study of α-crystallin in vitro led us to the assumption that, of the two α-crystallin isoforms (αA- and αB-crystallins), it is αA-crystallin that plays the role of a special chaperone for αB-crystallin. In addition, our data on X-ray diffraction analysis of α-crystallin at the sample concentration of about 170–190 mg/mL allowed us to assume that, at a high concentration, the eye lens α-crystallin can be in a gel-like stage. Finally, we conclude that, since all the accumulated data on structural-functional studies of α-crystallin were carried out under conditions far from native, they cannot adequately reflect the features of the functioning of α-crystallin in vivo.

## 1. Introduction

α-Crystallin is the main structural protein in the eye lens and makes up about 40% of its dry weight. In addition to α-crystallin, the lens contains β- and γ-crystallins, which together comprise 90% of all proteins in the fibroid cells forming the lens [1,2,3,4,5,6]. The concentration of α-crystallin in the eye lens is really high, reaching 450 mg/mL in humans and as high as 1100 mg/mL in fish [1,2,3,4,5,6,7]. α-Crystallin is responsible for the transparency of the lens, creating the required refractive index (n = 1.4 in human eye lens; for comparison, n = 1.33 in water) [8]. α-Crystallin consists of αA- and αB-crystallins, the identity between the two isoforms in mammalians is currently established as only 35%, much lower than the previously published figure of 60% [9,10,11]. αА- and αВ-crystallins consist of 173 and 175 amino acid residues and have a molecular mass of 19.8 and 20.0 kDa, respectively [8,12]. αА- and αВ-crytallins in humans are encoded on different chromosomes: 21 and 11, respectively [13,14]. αА-Crystallin (HSPB4) is mainly expressed in the eye lens, while αB-crystallin (HSPB5) is also expressed in other tissues, such as brain, heart, and muscles [15]. αB-Crystallin is associated with myopathies; neurodegenerative disorders including Parkinson’s, Alzheimer’s, and Creutzfeldt–Jakob diseases [16,17]; various sclerotic [18] and oncologic diseases [19,20]; as well as apoptosis [21], which emphasizes its importance for cellular proteostasis. αB-Crystallin in vitro prevents stress-induced aggregation of partially folded polypeptides [10]. The list of processes involving αB-crystallins continues to grow. In the eye lens, αA- and αB-isoforms form hetero (αA/αB) oligomers [22,23]. In the lens of most vertebrates, the typical molar ratio of the αA- to αB-forms is 3:1 [24], but it can reach 19:1 [25].

A hetero-oligomeric mixture of the two isoforms cannot be separated under non-denaturing conditions [26]. It was shown that both α-crystallin and its isoforms form αА- and αВ-oligomeric complexes [23,27], and all of them are somewhat heterogeneous.

In addition to the structural role, both α-crystallin and its αА- and αВ-isoforms were shown to perform a chaperone-like function and to belong to the family of small heat-shock proteins (sHsp) [28,29,30]. In mammals, proteins of this family are expressed in response to different physiological stresses (temperature and pH) and prevent protein denaturation, which may lead to aggregation [31,32]. The chaperone-like function of α-crystallin prevents the formation of large light-scattering aggregates, thus disrupting the mechanism, which would otherwise lead to the development of cataract [32]. α-Crystallin suppresses the aggregation under denaturing conditions of β-, γ-crystallins and in a wide range of conditions [9,30,33]. It was demonstrated that, under the physiological conditions, both the homopolymers and heteropolymers have a chaperone-like activity [8].

Like other proteins of the sHsps family, αА- and αВ-monomers contain three distinct regions: (a) the least conserved N-terminal domain (fragment of about 60 amino acid residues) [11]; (b) a structurally conserved central α-crystallin domain (ACD) of about 90 amino acid residues; and (c) an intermediately conserved C-terminal region of about 25 amino acid residues [9]. According to the published data, all three regions of the two α-crystallin isoforms are involved in the substrate recognition and binding, and thus, all three are involved in the chaperone activity [23].

It was shown that α-crystallin forms polydisperse oligomer complexes, this property impeding the obtaining of its crystals and resolving its molecular structure by using X-ray crystallography. At the same time, this property seems to represent an indispensable adaptation, which has evolved to prevent the possibility of crystallization of α-crystallin at its exemplary high concentration in the eye lens. 

The crystallin structure was determined for some members of the sHsp protein family, such as sHsp 16.5 from *Methanococcus jannaschii* [34]. The sHsp 16.5 complex is always an octahedral symmetry oligomer of 24 subunits, possessing an internal cavity. The sHsp 16.9 complex represents a dodecameric double disc with 32 symmetry and a cavity inside. Using the cryo-EM data [35], it was shown that the recombinant human αB-crystallin forms the most homogeneous oligomers. To analyze the images, particles with an average molecular mass of 650 kDa, which formed an oligomer consisting of about 32 subunits, were chosen. For such particles of human αB-crystallin, a low-resolution model (34 Å) exhibiting a roughly spherical morphology of 8–18 nm in diameter and a large central cavity of 3–10 nm in diameter was constructed [35]. In the latter study, native bovine α-crystallin was also analyzed. Natural (authentic) α-crystallin, extracted from the bovine eye lens, forms asymmetric oligomers but with a larger diameter compared to that of the recombinant human αB-crystallin and with higher heterogeneity [28,35]. The cryo-EM data allowed obtaining a model of the natural α-crystallin at a resolution of about 44 Å; it was noted that, in contrast to the recombinant αB-crystallin, the α-crystallin oligomer is a rounded particle without a distinct cavity, since only a slight decrease in density is observed in the central parts of the complexes [27,28]. Size-exclusion chromatography, analytical ultracentrifugation, as well as transmission electron microscopy (negative staining) were used to analyze the structures of recombinant αA- and αB-crystallins and to compare those with the structure of authentic (native) α-crystallin purified from bovine eye lens (αL-crystallin) [27]. According to the sedimentation data, αB-crystallin is the most homogeneous (16.5 S) while αL-crystallin (20 S and higher) is the most heterogeneous. Since αB-crystallin has a slight polydispersity, Peschek et al. analyzed the quaternary structure of its oligomeric complex and constructed a model at about 20 Å resolution [27]. The model of the αB-crystallin complex was revised by Jehle et al. [36]; Peschek et al. further studied the αB-crystallin complex with regard of its chaperone function [37]. Members of the sHsps family possess no ATPase activity; therefore, their chaperone function is regulated in a different way. For many mammalian sHsps, phosphorylation was shown to play the principal role here [37]. In this regard, it is of interest to study changes in the structure of the αB-crystallin complex during its phosphorylation. By combining different experimental approaches, Peschek et al. showed that the destabilization of intersubunit interactions induced by phosphorylation via the *N*-terminal domain leads to rearrangement of the oligomer ensemble and the enlargement of smaller active complexes, predominantly 12-mers and 6-mers [37]. It was noted that a hexamer is the main building block of αB-crystallin, which can form complexes consisting of up to 48 subunits, while an additional building block is a dimer. These particular small complexes are associated with an increased chaperone activity in vivo and in vitro and more effective interaction with proteins, the dimer exhibiting the highest chaperone activity [37].

Thus, currently available is a model of the α-crystallin complex with a resolution of only about 44 Å [28], while no further attempts have been made to study the native structure owing to the high heterogeneity of the complex. Recently the different reduced models of αA-crystallin complexes were proposed [38]. The absence of a correct model complicates interpretation of the mechanism of chaperone activity of both native α-crystallin and αA-crystallin complexes. 

In the past ten years, a lot of experimental data on the functioning of α-crystallin and its isoforms have been accumulated, summarized in the following reviews [8,10,23]. Despite intensive studies of the α-crystallin complex and the use of new modern research methods to study the behavior of α-crystallin complex, the accumulation of quantitative data continues and many questions remain regarding its structural features and functioning. Due to the acceleration of technological progress and the introduction of an increasing number of modern computer technologies, the load on the organs of vision increases significantly, and eye diseases cover more and more age groups. Cataract, one of the reasons for the processes of aggregation of α-crystallin, covers more and more populations; this disease is becoming younger every year. Due to the uniqueness of α-crystallin and, in particular, its presence in the eye lens at an extremely high concentration, the reliability of all accumulated in vivo data and their correlation with in vitro conditions is becoming increasingly important. Finding answers to these questions will determine the strategy for developing effective therapeutic drugs for the prevention and treatment of eye lens disorders, one of which is cataract.

In this regard, it is important to continue research using the data already obtained. Analysis of this data will help to move from a quantitative accumulation of the results achieved to a qualitative leap in understanding the functioning of α-crystallin. Despite the fact that methods for studying proteins at extremely high concentrations have not yet been developed, it is now possible to use the accumulated data for coming to certain conclusions and to put forward a number of assumptions on the work of α-crystallin in vivo.

Herein, we analyzed the native α-crystallin complex from the central (nucleus) and peripheral (cortex) parts of eye lens using the EM method (negative staining) and X-ray analysis. We also used the data of some previous studies, and based on them, we propose some new ideas related to the functioning of α-crystallin in eye lens and the possibility of correlating in vivo results with in vitro conditions.

## 2. Materials and Methods 

### 2.1. Samples

All reagents were obtained from Sigma-Aldrich (Westport Center, USA): TRIS hydrochloride (Tris-HCl), NaCl, and Ethylenediaminetetraacetic acid (EDTA).

We used α-crystallin preparations from the cortex and nucleus of bovine eye lens isolated from lenses of 2-year-old steers and purified as previously described [39,40]. The samples were presented by K.O. Muranov (see Acknowledgements). To isolate α-crystallin from the nucleus, the material from the central part of eye lens was used. The isolated preparations were compared to the preparation produced by Sigma-Aldrich (lot#SLBD1923V). All methods were carried out in accordance with relevant guidelines and regulations. The purity of the α-crystallin preparations was tested using standard SDS gel-electrophoresis.

### 2.2. Electron Microscopy and Negative Staining

Prior to EM experiments, the preparation of α-crystallin (Sigma-Aldrich) was dissolved at 5–20 mg/mL in the buffer containing 20 mM Tris-HCl (pH 7.5), 100 mM NaCl, and 1 mM EDTA and dialyzed against the same buffer overnight at 4 °C. Before making samples for EM, preparations were centrifuged for 1 h at 16.5 rpm at 4 °C to remove large aggregates. The supernatant was taken, and the concentration was adjusted to 0.2 mg/mL and incubated for 30 min at 37 °C. EM samples were prepared according to the negative staining method, either at 4 °C or after heating at 37 °C for 30 min. A copper grid (400 mesh) coated with a formvar film (0.2%) was mounted on a sample drop (10 μL). After 5-min absorption, the grid was negatively stained for 1.5–2.0 min with 1% (weight/volume) aqueous solution of uranyl acetate. The excess of the staining agent was removed with filter paper. The preparations were analyzed using a JEM-100С (JEOL, Tokyo, Japan) transmission electron microscope at the accelerating voltage of 80 kV. Images were recorded on to Kodak electron image film (SO-163) (Kodak Electron Image Film, New York, USA) at nominal magnification of 40,000. 

To form a gel (at 170 mg/mL), the α-crystallin preparation was concentrated at room temperature using an Eppendorf 5301 vacuum concentrator (Eppendorf, Hamburg, Germany). The α-crystallin gel was prepared for EM analysis as follows. A small piece of gel-like material (about 2–3 μL) was pipetted into a buffer (about 20 μL) with a spout and mixed by pipetting in and out. The preparation (about 5 μL) was placed on a mesh with formvar film. After adsorption of the preparation (1 min), the mesh was washed on a drop of buffer (50 μL) for 30 s. Contrasting with uranyl acetate, preparation was carried out as described above.

The use of Atom Force Microscopy (AFM) in the study of highly heterogeneous complexes is irrational. We will observe a large number of particles of different sizes, and at the same time, the “resolution” of even the most frequently occurring particles with the same parameters will be worse than using EM. Moreover, this is due to not only heterogeneity but also the insufficient density of the complexes. When using AFM, we will observe rounded particles with fuzzy edges. The use of AFM is well suited for the study of more stringent and homogeneous preparations. AFM showed itself well in studying the growth dynamics of various fibrillar structures when measured in liquid.

### 2.3. X-ray Analysis

For X-ray analysis, the sample of α-crystallin from the cortex was prepared at the concentration of 19 mg/mL in buffer containing 20 mM Tris-HCl (pH 7.5), 100 mM NaCl, and 1 mM EDTA. The sample was centrifuged at 16 rpm for 1 h at 4 °C to precipitate large aggregates; the supernatant was removed and heated for 1 h at 37 °C. The sample was concentrated to 170–190 mg/mL at room temperature using an Eppendorf 5301 vacuum concentrator before gel formation (about 2 h). A sample drop of about 6 μl was placed in the space (about 1 mm) between the tips of two glass rods with the diameter of about 1 mm, coated with bee wax. When the preparation was lyophilized (about 2–3 h in air), the preparation rod was 1 mm long and its diameter was about 0.1 mm. X-ray diffraction patterns were obtained using an X8 Proteum System (“Bruker AXS”, Karlsruhe, Germany) and Cu Kα radiation (λ = 1.54 Å). To position the sample at the correct angle to the X-ray beam, a 4-circle Kappa-goniometer (“HUBER”, Rimsting, Germany) was used.

### 2.4. Calculation of Occupancy of a Unit Volume by α-Crystallin

To estimate the volume occupied by complexes and monomers at a given concentration, we consider the following parameters: V_сomplex_ = 1552 nm^3^, M_сomplex_ = 837 kDa (about 42 monomers) (according to Siezen and Berger), V_monomer_ (accessible volume) = 20 nm^3^, and M_monomer_ = 20 kDa. Moreover, if you make a calculation for a concentration of 500 mg/mL and take into account that 1l = 10^24^ nm^3^, it turns out that the monomers will fill 0.3V_0_ (cell or 1 L) and complexes 0.55 × V_0_ (cell or 1 L). Among all packages of balls of equal size in three-dimensional space, the greatest average density is hexagonal close packing, k = V_spheres_/V_space_ = ≈0.74 (the Kepler conjecture says that this is the best that can be done). With random packing, this coefficient is 0.65. This means that the volume occupied by small balls (monomers) will be 0.45 × V_0_, and in the case of oligomers, it will already be 0.83 of the volume under consideration. When the concentration doubles to 1000 mg/mL, the monomers will occupy 0.9 of the volume under consideration and the complexes will no longer fit in the cell (1.66 × V_0_).

## 3. Results 

### 3.1. Electron Microscopy Analysis of α-Crystallin

The quality of our preparations of bovine α-crystallin isolated as described [39,40] as well as those produced by Sigma-Aldrich was analyzed using denaturing SDS gel electrophoresis (Appendix A). All preparations were found to be at least 95% pure (Appendix A). The EM analysis revealed that the preparations looked identical (Figure 1). In our further studies, we used natural α-crystallin isolated from the bovine eye lens.

The obtained EM data showed that the α-crystallin preparation represented heterogeneous oligomer complexes of different dimensions (from 12 to 25 nm in diameter) (Figure 2). 

In the preparation field (Figure 2A), some distinct complexes of α-crystallin are seen, their diameters vary, and they are stained differently by the uranyl acetate. The particles with a diameter of about 12–14 nm are stained most intensively, which is an indirect evidence that their homogeneity and compactness are higher. It is also visible that the larger the particle’s diameter, the lighter it is stained. This may be indicative of lower compactness and orderliness of the larger oligomers and/or their lower height. Heterogeneity of the oligomer complexes of α-crystallin was demonstrated in the set of individual particles in Figure 2B. It can be easily seen that some oligomers are assembled into aggregates (the open arrow in the center of Figure 2A) while others interact with each other, forming some elongated particles (the closed arrow). Analogous aggregates have been mentioned earlier [27,41]. In aggregates, the particles have a diameter of about 11–12 nm, which is smaller than the average diameter of 12–14 nm. This might have been an artifact of the staining of higher-order complexes with uranyl acetate and/or of possible compactization of complexes upon their interaction. It should be noted that a great number of small particles of different dimensions can be seen; they are poorly stained and consequently were not analyzed. For comparison, one can see EM data for the recombinant αB-crystallin from bovine eye lens (Appendix A).

The distribution of oligomer complexes of α-crystallin with the diameter of 12–25 nm (particles from one EM film were calculated) is presented in Table 1. Smaller particles and oligomer complexes forming large aggregates were not considered. From 1143 oligomer particles (analysis of one EM film) of a negatively stained preparation of native α-crystallin from bovine eye lens, more than 50% have a diameter of about 12–14 nm. For further analysis, we selected particles with a diameter of about 12–14 nm. Such particles are the most abundant and exhibit better staining. Figure 2C highlights the 12–14-nm particles used for 3D reconstruction (Appendix A).

### 3.2. Comparison of α-Crystallin from Cortex and Nuclear Region

We also compared α-crystallin from the cortex and from the nuclear region of eye lens (Figure 3). According to the EM analysis data, the α-crystallin complexes from both the nuclear region and cortex of eye lens are significantly heterogeneous. EM images of α-crystallin from the cortex show rounded particles with diameters of 12–25 nm (most particles have the diameter of about 13.5 nm) (Figure 2), while the image of that isolated from the nucleus shows the bean-shaped particles with diameter of 15–20 nm (the average width of about 15 nm and height about 20 nm) (Figure 3B). Thus, in addition to the increased size of the particles of the α-crystallin from the nucleus, a change in their morphology is observed: the particles become bean-shaped. In both cases, a large number of smaller particles are seen in the field. They are poorly stained and not observed with ease. Evidence that α-crystallins from the nucleus and cortex have a different molecular weight and sedimentation coefficient but that it is the same protein has been known since 1973 [42].

### 3.3. Three Dimensional (3D) Reconstruction

We attempted to develop a 3D model of the α-crystallin complex from the cortical part of eye lens because a model of a native α-crystallin complex from bovine eye lens was previously published [28]. Particles of 13.5 nm in diameter were selected for analysis. They were well stained with uranyl acetate and made up more than 50% of all observed complexes. According to the published data, native α-crystallin from the bovine eye lens forms heterogeneous complexes with the molecular mass between 300 kDa to 1200 kDa [35]. As was shown earlier by the sedimentation analysis, oligomeric complexes of native α-crystallin have the sedimentation coefficient between 15 S to 45 S [27]. However, the dominant fraction of α-crystallin exhibits the sedimentation coefficient of 20 S [27]. The presence of a large number of peaks (populations) of α-crystallin particles with sedimentation coefficients of 18.3–30.0 S and higher was also reported [43]. In this case, most of the particles had sedimentation coefficients of about 22 S. The relationship between the logarithm of the sedimentation coefficient and the molecular mass of typical globular proteins as well as protein complexes yields the molecular mass of the α-crystallin complex of about 740 kDa with a sedimentation coefficient of 20–22 S (Figure 4). This value of the molecular mass correlates well with the experimental value for the α-crystallin complex [44]. We constructed a 3D model for the particles of the α-crystallin complex with the diameter of 13.5 nm and with the mass of about 650 kDa at a resolution of about 20 Å (Appendix A). Such particles exhibit a rounded, slightly elongated shape without a distinct inner cavity, which agrees well with the previously published data [28]. Attempts to construct a model for the populations of particles with larger sizes and molecular weights (700, 750, and 800 kDa) gave a similar result: the rounded, slightly bean-shaped particles of somewhat larger dimensions were obtained. However, invariably, there was no noticeable cavity inside.

It was shown that the complex model (αB-crystallin/lactalbumin) with a resolution of 42 Å (cryo-EM) did not show a pronounced cavity inside, but a noticeable increase in the size of the complex and the heterogeneity of the complexes was observed [28].

It is interesting to note that the heterogeneity of complexes facilitates the binding of a heterogeneous population of target proteins. The English team [46] investigated the binding of α-crystallin to γE-crystallin using small-angle neutron scattering and noted that, at a stress temperature (65 °C), structural changes of the αB-crystallin complex take place. The authors assumed that, as in the case with GroEL, when interacting with the protein, the origin or penetration of the target unfolded protein into the cavity of the complex or the interaction of the protein with the surface of the complex occurs. That is, two mechanisms for the functioning of α-crystallin as a chaperone are proposed. However, while for small proteins/peptides it is still possible to assume the mechanism of penetration of the denatured protein into the chaperone structure, for proteins with a molecular mass of 40–60 kDa, this is already problematic. A model of the 24-subunit αB-crystallin complex with a resolution of about 20 Å was presented in Reference [27]. The diameter of the complex is about 13.5 nm, the entrance to the cavity is about 3.5 nm, and its inner diameter is about 8.5 nm. When the model was refined in 2013 [37], it was assumed that the number of subunits of the complex can be from 24 to 48. In this case, the density of particles increases and penetration of the unfolded protein into the complex and its localization there becomes problematic. It seems to us that the existence of two mechanisms of the α-crystallin chaperone activity for proteins with different molecular masses (with penetration of the target protein into the complex or interaction with its surface) is not justified. From the point of view of nature (evolution), there should exist a universal mechanism for the functioning of the α-crystallin complex as a chaperone. It seems to us that such a mechanism may be the dissociation of the complex into smaller aggregates and their binding to the unfolded target protein.

### 3.4. X-ray Diffraction Analysis

To prepare a sample of α-crystallin (the nucleus) for X-ray analysis, its concentration was increased to 170–190 mg/mL (see the Material and Methods section). At this concentration, the preparation turns into a gel, but it is still possible to place it as a drop between the tips of two glass rods. As shown by X-ray diffraction analysis, the α-crystallin preparation has reflection characteristics of amyloid fibrils: meridional of 4.6 Å and equatorial of 9.2 Å (Figure 5). It can be concluded that, upon concentrating, the preparation forms fibrils. The EM analysis of the gel-like preparation of α-crystallin (Figure 6A) demonstrated that, at the concentration of 190 mg/mL, α-crystallin forms dense aggregates of heterogeneous oligomer particles of 14–18 nm (Figure 6B). However, in less dense areas, it can be seen that α-crystallin oligomers tend to align in long formations resembling short fibrils (Figure 6C). Separate oligomer complexes with parameters characteristic of α-crystallin from the nucleus can be observed in areas where the preparation was not adsorbed (less adsorption occurred) (Figure 2, Figure 3, and Figure 6D). Amyloidogenic fibrils of α-crystallin have been described previously [47,48]. However, for their formation, α-crystallin was treated with 1 M guanidine hydrochloride [47,48], i.e., destabilization of α-crystallin was used.

The data of X-ray diffraction analysis of the gel-concentrated preparation show that α-crystallin has the diffraction pattern characteristic of amyloid structures (Figure 6C). However, according to the data of EM analysis, the preparation of α-crystallin is predominantly densely packed heterogeneous oligomer complexes. α-Crystallin is known to have β-structured regions that form small β-sheets, as shown for the ACD domain (pdb file 2klr). When any amyloidogenic proteins or peptides are prepared for X-ray diffraction analysis, the concentration of preparations increases (drying of a sample drop; see the technique of sample preparation for X-ray diffraction analysis). With prolonged incubation and adequate concentration, many amyloid proteins/peptides form large clusters of fibrils, which can turn into a gel (our unpublished date; see Appendix A) [50]. The concentration at which gels are formed depends on the amino acid sequence and is usually 5–20 mg/mL. In the case of α-crystallin, as shown by our data, the gel-like mass is formed at C = 170–190 mg/mL. In all cases of sample preparation of amyloidogenic proteins/peptides for X-ray diffraction analysis, we analyzed the preparations of lyophilized gels. Based on the data of X-ray and EM analysis, we propose that α-crystallin in eye lens can have a gel-like form; that is, it represents a biogel.

The β- and γ-crystallins interact with α-crystallin and, among other things, can contribute to increasing its heterogeneity. Strengthening the heterogeneity of α-crystallin at an extremely high concentration, especially in the nuclear region of the eye lens, prevents its crystallization. As a result, at a high concentration of α-crystallin in the eye lens under in vivo conditions, α-crystallin can be in a gel-like (amorphous) state.

## 4. Discussion

α-Crystallin represents highly heterogeneous oligomer complexes with molecular mass from 300 to 1200 kDa and average molecular mass of about 700 kDa. This distribution of molecular mass is defined by many parameters, including temperature, concentration, ionic strength, extraction method, eye lens age, and source [10]. It should be noted that using such methods as size-exclusion chromatography, analytical ultracentrifugation, small-angle X-ray scattering, dynamic light scattering, etc., the different research teams obtained somewhat mismatching values of both the molecular masses and the sedimentation coefficients for the peak fractions [10,27,43,44,51,52]. The simultaneous use of size-exclusion chromatography, light scattering, and estimation of the refractive index allowed the researchers to determine the molecular mass of the peak fraction of α-crystallin for the natural bovine α-crystallin complex to be as high as 700 kDa, with the ratio of αA- to αB-crystallins being 3:1 [10]. We used literature data for the average (22.5 S) peak sedimentation coefficient, and our theoretical calculations showed that this value corresponds to the particle with compact packing and molecular mass of about 740 kDa, which is compatible with the experimentally obtained value (Appendix A).

The most accurate determination of molecular mass is required for designing a model of any complex, taking into account the structure/function relationship in proteins. Therefore, numerous studies are aimed at determining the structure of such an important protein complex as α-crystallin. Due to the high heterogeneity of the α-crystallin complex and, therefore, its inaccessibility in the crystalline form, EM methods have become the most successful ones in this field. The generally improved research tools, cryo-electron microscopy, and the enhanced computer processing of individual images have recently allowed to obtain the first models of both the authentic (natural) α-crystallin complex isolated from the bovine eye lens and the recombinant αB-crystallin complex [27,28,36,37,38]. The studies of the αB-crystallin complex using negative staining in combination with computer processing of individual particles made it possible to improve the model of the αB-crystallin complex (resolution about 20 Å) [27,36,37]. Nevertheless, little progress was achieved, except for the creation of one model [28], with highly heterogeneous authentic α-crystallin and αA-recombinant complexes [28,38]. So far, only the model of bovine α-crystallin complex has been proposed at the resolution of 44 Å. Our model of the largest population of the α-crystallin complex (with the diameter of particles of about 13.5 nm and molecular mass of 650 kDa at the resolution of about 20 Å) demonstrated that the 3D structure of the complex did not differ greatly from that proposed earlier [28]. We believe that, due to a high heterogeneous nature of the α-crystallin complex, it is impossible to determine its structure reliably. In this connection, pinpointing is the research [52] performed using the EM method to analyze α-crystallin preparations from different fractions upon isolation of α-crystallin. The EM data show that the diameter of particles varies from 13.5 nm to 16.0 nm. The morphology of all particles from different fractions is similar (rounded slightly ellipsoidal particles); only the dimensions of the complexes differ. As the same authors reported [52,53], according to the data of sedimentation analysis and small-angle X-ray scattering, the average molecular mass for such particles should be 19.2 S (850 kDa according to the authors of the papers). 

Literature reviews devoted to examination of α-crystallin reveal that many issues related to the structure and functioning of α-crystallin remained unanswered and that various experimental data allow making notable conclusions. Firstly, the adaptation eliminates the risk of crystallization of α-crystallin at its unusually high concentration in the eye lens, thus safeguarding the lens transparency. Secondly, it can also be associated with a function similar to a chaperone. Therefore, we suggest that oligomerization of α-crystallin is another evolutionary adaptation allowing retention of an inactive α-crystallin form. This state of α-crystallin enables a quick response to various changes in the physiological conditions (stress, pH, temperature, modifications, etc.), allowing the chaperone resource to be used according to the situation. Therefore, it should be part of the α-crystallin functioning, since the complex dissociates into smaller oligomers up to dimers that have a higher chaperone activity. This model of behavior of the α-crystallin complex is supported by numerous experimental data and assumptions. For example, phosphorylation of the α-crystallin complex causes its dissociation into dimers with significantly higher chaperone activity [37]. In addition, according to the data of Reference [54], with an increase in temperature, sHsp 16.3 from *Mycobacterium tuberculosis* dissociates and exhibits higher chaperone activity. It is also asserted that dissociation is necessary for the implementation of a function similar to chaperone. It was concluded that the ambient temperature can trigger the chaperone activity. In addition, it is assumed that oligomerization and the substrate binding are two competing processes and that temperature plays a regulatory role. 

It has been shown that, although αB-crystallin prevents aggregation of a large number of proteins not only in the eye lens but also in some other organs as well, αA-crystallin is found almost exclusively in the eye lens [10,23]. This might provide evidence for significant specialization of αA-crystallin as a “personal” chaperone for the eye lens crystallin. This is indirectly supported by the fact that, with age, the ratio of αA- and αB-crystallins in α-crystallin changes and that the proportion of αA-crystallin decreases to 3:2 [55]. For αA-crystallin gene knockout, cataracts develop in mice, beginning at the core of the lens and spreading with age to the peripheral cortical part. In addition, the formation of inclusion bodies with aggregates of αB- and γ-crystallins was observed [10,56]. That is, αA-crystallin is required to maintain the transparency of the eye and to prevent aggregation of αB-crystallin. At the same time, knockout on the αB-crystallin gene showed that αB-crystallin is not necessary to maintain the transparency of the lens in the early stages of life but that, at the same time, knockout on the αB-crystallin gene dramatically reduces life duration [10,56].

At present, there are no reliable methods for studying proteins at high concentrations. Therefore, the eye lens remains one of the most puzzling and poorly studied objects. We considered that a lot of current data should be reexamined regarding the modeling under in vivo conditions. A simple calculation of the α-crystallin packing within complexes at its concentration of about 500 mg/mL per unit volume shows that, as we observed in vivo for human eye lens, this concentration is critical because α-crystallin complexes fill 0.83 the entire cell volume. Taking into account that the eye cell contains also other crystallins as well as cytoskeleton proteins, it becomes evident that, under in vivo conditions, α-crystallin does not form large complexes because, when the same volume is occupied by monomers or smaller aggregates, enough space still remains (about 0.55 the entire volume) to accommodate all other components of the eye lens fiber cells (see Section 2.4. Calculation of occupancy of a unit volume by α-crystallin and Figure 7). If we try to place α-crystallin complexes at concentration of 1000 mg/mL (as in some fish) in the cell, it will occupy a larger volume than the cell itself (1.66 the entire volume). Only if α-crystallin will have the form of smaller complexes (up to dimers) it will fill the cell so that there will remain space for other crystallins and the cytoskeleton proteins, and in this case, the chaperone activity will even increase.

This supposition is supported by the fact that αB-crystallin outside the eye lens does not form complexes and that, when exhibiting its chaperone functions, it is involved in intricate interactions with some other sHsp proteins [57,58,59]. Moreover, the αA- and αB-genes are localized on different chromosomes, and during synthesis, they are expressed at different times and in different eye lens cells, allowing us to conclude that they exist quite independently of each other [60,61,62]. Thus, a question remains about the behavior of α-crystallin in vitro vs. in vivo. Under native conditions, the concentration of α-crystallins differs greatly. The high concentration of α-crystallin decelerates the exchange between the oligomer complexes (the crowding effect), which is observed under discharge conditions in the experiments in vitro when free diffusion of both particles and their components takes place [23]. Taking into account the accumulated data, we support the authors [23] advising to more thoughtfully analyze the data on the chaperone activity of the eye lens α-crystallins and to focus special attention on the roles of the αA- and αB-isoforms. It is probable that the key mechanism in the functioning of αA- and αB-crystallins in the eye lens is also phosphorylation, which triggers dissociation of oligomers into smaller aggregates and increases the chaperone activity [23].

The analysis of the literature allows to assume that polydispersity of α-crystallin is not only an indispensable evolutionary adaptation to prevent crystallization of the eye lens α-crystallin but also a wasteful use of α-crystallin throughout lifetime. This suggests that oligomeric α-crystallin is an inactive form of α-crystallins that form oligomers, and with aging as well as during various stresses, the complex dissociates into smaller aggregates, up to dimers, which have higher chaperone activity in this state. However, this may indicate that, under in vivo conditions, complexes of such dimensions cannot be even formed. However, the principal conclusion of our research is that, despite the intensive studies of such a vital object as the eye lens, there are still more questions than answers. So far, current studies cannot yield a qualitatively new assessment of the α-crystallin functioning in vivo. This is explained by the fact that the concentration of α-crystallin in the eye lens is excessively high, and there are no reliable methods for studying such concentrations at present.

As shown, α-crystallin fibrils exhibit chaperone activity; so, both dimers and α-crystallin fibrils exhibit chaperone activity [48]. Current data suggest that, at a high α-crystallin concentration in the eye lens, α-crystallin has a gel-like state and, even in the form of amyloids, retains the chaperone function [48]. We assume that α-crystallin at its high concentration in the eye lens exists in the form of a gel (biogel); that is, the eye lens is a gel-like state of α-crystallin. This is an indirect evidence that the chaperone activity of α-crystallin is displayed completely during dissociation of α-crystallin complexes, and the complexes are a form of preserving α-crystallin in a less active state until potential stress situations arise. 

## 5. Conclusions

In connection with the rapid development of new research methods and improvements of traditional ones, more and more data are accumulating on the structural and functional studies of the α-crystallin complex. Based on our own and analysis of literary data, we came to the following conclusions:Both the heterogeneity of α-crystallin complexes and the fact of their formation represent some evolutionary adaptations. At high concentrations of α-crystallin in the eye lens, the heterogeneity of the α-crystallin complexes prevents protein crystallization. An increase in the activity of chaperone during the dissociation of complexes suggests that α-crystallin remains inactive in these complexes, which allows its discrete use under stress.It is impossible to obtain a reliable model of α-crystallin.αA-Crystallin is likely a specific lens chaperone.At a very high concentration, α-crystallin in eye lens can have a gel-like state; that is, the eye lens is a biogel.In vivo, due to the molecular crowding conditions, everything may be quite different from in vitro conditions, which leads to the conclusion that a more thorough analysis of the accumulated data is necessary.

## Figures and Tables

**Figure 1 biology-09-00085-f001:**
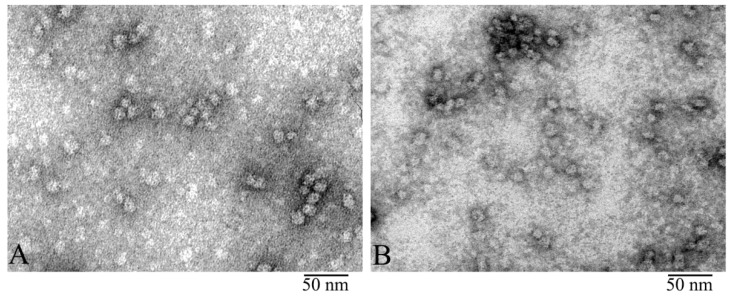
Electron microscopy (EM) of negatively stained α-crystallin from bovine eye lens: Comparison of natural α-crystallins (**A**) and the preparation produced by Sigma (**B**). EM images show that both preparations form heterogeneous complexes identical in shape and dimensions. Preparation conditions for EM analysis: C = 0.2 mg/ ml, incubation for 30 min at 37 °C in 20 mM Tris-HCl buffer (pH 7.5), 100 mM NaCl, and 1 mM EDTA.

**Figure 2 biology-09-00085-f002:**
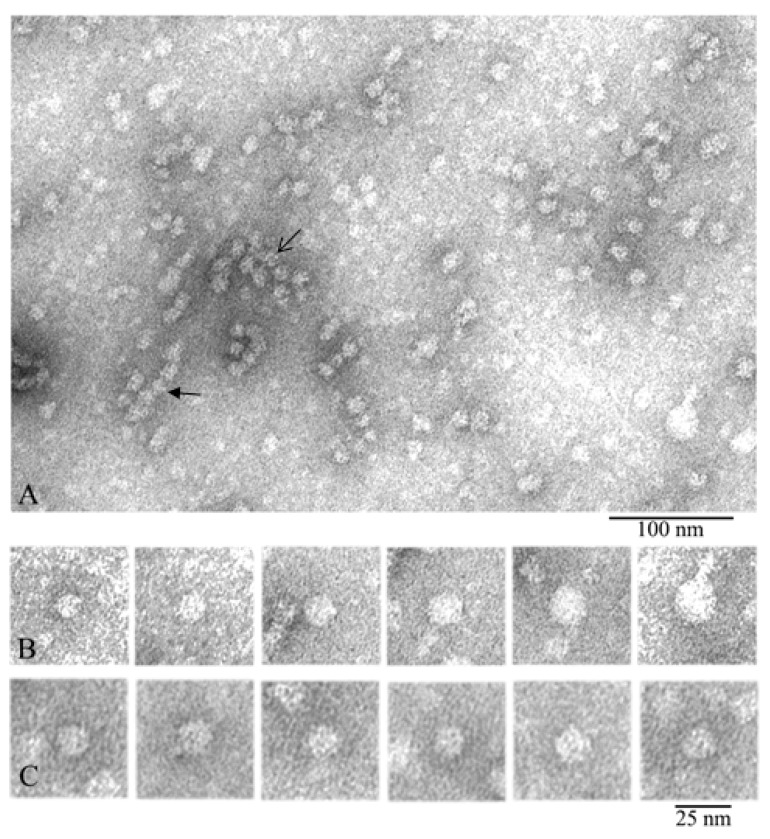
Electron microscopy of negatively stained natural α-crystallin from bovine eye lens. (**A**) Preparation field: the open arrow points to an oligomer aggregate, and the closed arrow indicates a stack of 4–5 particles. (**B**) Representative images of oligomer complexes of different sizes from relatively small to large. (**C**) Representative images of 12–14-nm oligomers from the large population (used for 3D reconstruction). Preparation conditions for EM analysis: C = 0.2 mg/mL, incubation for 30 min at 37 °C in 20 mM Tris-HCl buffer (pH 7.5), 100 mM NaCl, and 1 mM EDTA.

**Figure 3 biology-09-00085-f003:**
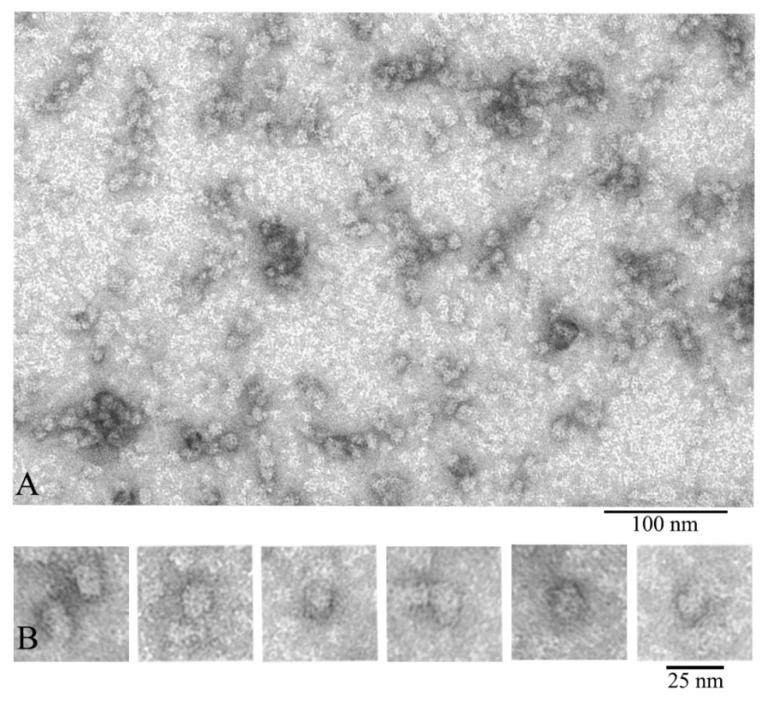
Electron microscopic images of the preparation of natural bovine α-crystallin from the eye lens nucleus: (**A**) preparation field; (**B**) a selection of single particles of the most representative population of the α-crystallin complex (asymmetric, bean-shaped particles with dimensions of about 15 × 20 nm). This population is represented by particles of a larger size compared to that of particles isolated from the cortex. Preparation conditions for EM analysis: C = 0.2 mg/mL, incubation for 30 min at 37 °C in 20 mM Tris-HCl buffer (pH 7.5), 100 mM NaCl, and 1 mM EDTA.

**Figure 4 biology-09-00085-f004:**
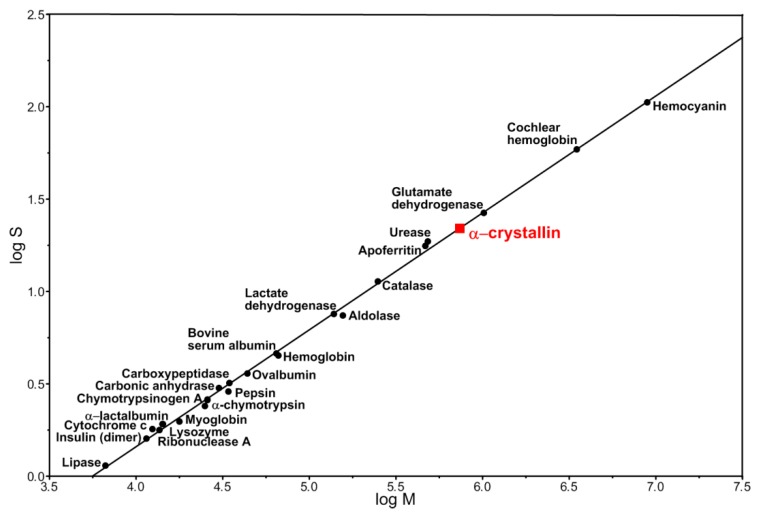
Logarithmic relationship between the sedimentation coefficient and the molecular mass of proteins and protein complexes [45].

**Figure 5 biology-09-00085-f005:**
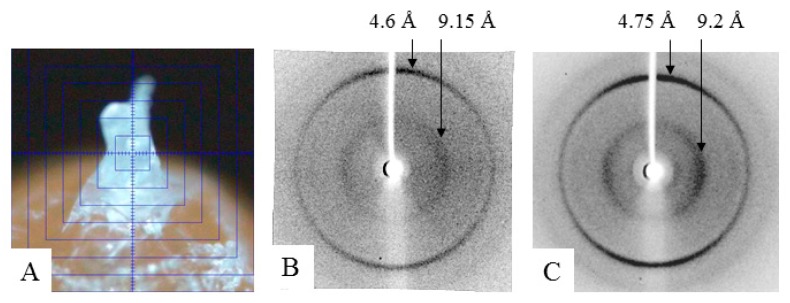
X-ray diffraction: (**A**) preparation rod of α-crystallin from bovine eye lens; (**B**) α-crystallin from bovine eye lens (nucleus); and (**C**) amyloidogenic fragment of protein Bgl2 from yeast cell wall [49].

**Figure 6 biology-09-00085-f006:**
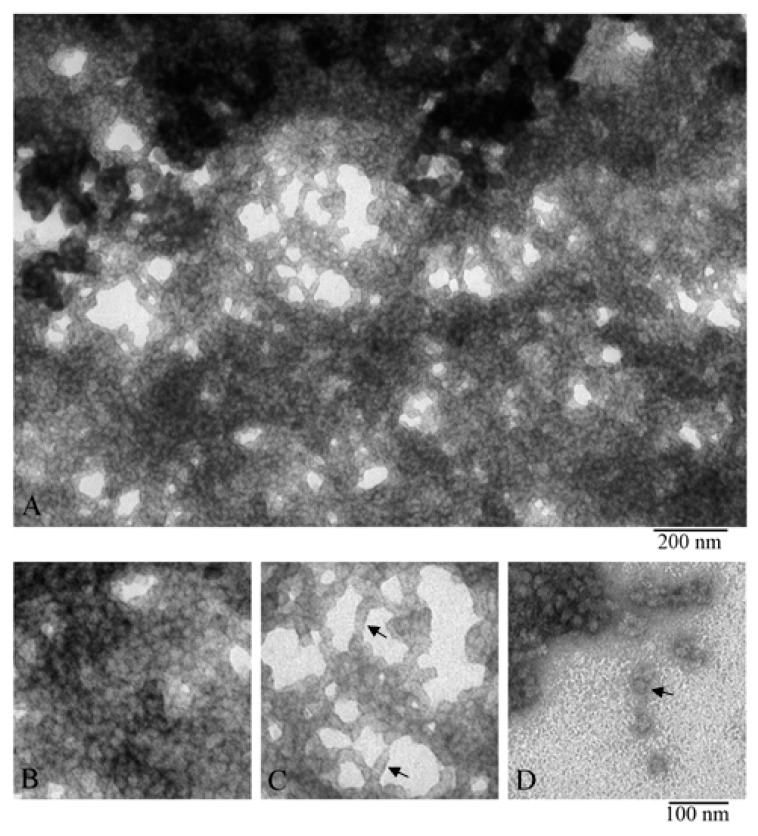
EM images of α-crystallin gel (nucleus). Fragments of the fields of α-crystallin gel: (**A,B**) field fragments with tight packing of heterogeneous oligomer complexes; (**C**) field fragment with lower density of α-crystallin complexes, arrows show short stacks of complexes; and (**D**) field fragment with single complexes (shown by the arrow).

**Figure 7 biology-09-00085-f007:**
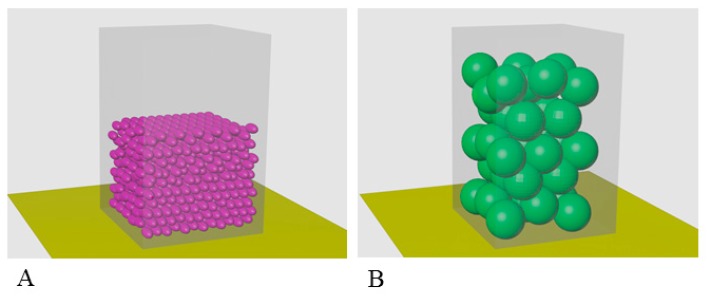
Model for filling the same volume with particles of α-crystallin of different sizes at the concentration of about 500 mg/mL: (**A**) monomeric α-crystallin; (**B**) α-crystallin complex. The calculation of the filling of the same volume with particles of different sizes shows that, at C = 500 mg/mL, the monomers will occupy about 0.45V (**A**) and that the complexes will occupy 0.83V (**B**). When the concentration of α-crystallin is up to 1000 mg/mL (as in some fish), α-crystallin in the form of complexes will take 1.66V, i.e., particles will not fit in the cell. Taking into account the fact that, in addition to α-crystallin, in the fibrous cells of the eye lens, there are β-, γ-crystallins and a number of other cellular proteins of the cytoskeleton, the existence of α-crystallin in the form of complexes is problematic.

**Table 1 biology-09-00085-t001:** Size distribution of native α-crystallin oligomer particles from bovine eye lens.

Complex Diameter (nm)	Number of Complexes	Portion (%)
12–14	605	53
15–19	309	27
20–25	229	20

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
