# Peer review of "Structural and Functional Peculiarities of α-Crystallin"

_biology, 2020, doi:10.3390/biology9040085_

Round 1

Reviewer 1 Report

The authors analyzed α-crystallin using Electron Microscopy and X-ray diffraction. They discussed the structure from the combination of their own data and also literary data. The authors suggested the structure of α-crystallin after long discussion. The aim of this study is not obvious, therefore the results and discussion are ambiguous in meaning. Their idea could be acceptable, however, I think it needs improving several points. It can be acceptable for publication after major and minor revisions.

Major revisions:

  1. Please declare the aim of this study in introduction and abstract. It is missing.
  2.  Please revise discussion and conclusions according to the aim of this study. Conclusion is also necessary in abstract.

Minor revisions:

  1. Page 4, if the authors conducted electron microscopy analysis of α-crystallin, image analysis should be done. Please add the results of size distribution of figure 1 and 2. Histogram or any type of figure is acceptable.
  2. Please give information for reagents and equipment used in this study, including Maker name, City and Country. Lot number in not necessary.
  3. Please add figure legend of supplementary figures.

Reviewer 2 Report

The manuscript by Selivanova and Galzitskaya uses electron microscopy and X-ray diffraction to propose a structural model for alpha-crystalline at 20A resolution based on the analysis of the most frequent population of oligomers found in the bovine eye lens. It further compares the morphology of alpha-crystalline complexes formed in different regions of the lens (specifically, in the nucleus and in the cortex). The topic is interesting and this result is important.

The article reads well. It is well written and it provides a good overview of the literature. However, the conclusions are not supported by the data. Specifically, the conclusions regarding the functional role of the alpha-crystalline as a chaperone are not supported by the data. They are instead speculations based on the authors’ reflections of their data and the literature. Since this is the most important part if this work for the target audience of this journal I have trouble in recommending the paper as it stands. I think that it would be more appropriate for a journal more specialized in structural biology.

I have two further recommendations. 1) The title is not adequate. It’s too long and lacks focus. 2) The major contribution of this manuscript, i.e. the proposed 3D model for the alpha-crystalline particles, is presented as SI and clearly it should be presented in the main text.

Finally, I was wondering if the authors could discuss the use of AFM as alternative/complementary method in the analysis of this type of protein complexes.

Reviewer 3 Report

The review by Selivanova and Galzitskaya entitled “A new view at the accepted idea of the work of α-crystallin. Own data and analysis of literary sources” aims to examine structure and function of the abundant lens protein α-crystallin.  The authors review the challenges of gaining structural insight into this polydisperse oligomeric target, the current structural models and the potential mechanisms of chaperone activity.  The paper also aims to propose some new insights based on research presented in this manuscript in which they develop their own model for the structure of α-crystallin using cryo-EM data, they make claims that the properties of α-crystallin in the lens (very high concentration and chaperone activity) are evolutionary adaptions, as well as suggesting α-crystallin in the lens exists as a biogel.  Understanding the α-crystallin structure and function is of broad to researchers in various fields, including cryo-EM, heat-shock proteins and those studying large oligomeric targets. 

Broad Comments

1.The authors do a good job in the introduction presenting the target of interest, its significance, the current models of the structure and the difficulties in obtaining reliable structural information.  I believe the one thing that could improve the introduction would be to incorporate the results of the recent αA-crystallin structure by Kaiser et al. (2019) Nat. Struct. Mol. Biol. 26, 1141-1150.

  1. Figure legends for the SI were missing.
  2. The methods section does not adequately describe how the authors made their model α-crystallin presented in Figure S2. Additionally, there are no methods presented for how it is determined that large oligomeric complexes and the high concentration of α-crystallin would fill the cell. It would be good to see how this was determined since it is offered as evidence of a biogel and evidence that only small dimers of α-crystallin can fit in the cell.
  3. Throughout the manuscript the authors mention that specific features of α-crystallin are evolutionary adaptations, but evolutionary argument is provided. Do α-crystallin exists in the eyes of simple species? From what did α-crystallin evolve from and what was its primitive role? There is no argument presented for this to be one of the main conclusions.
  4. The conclusion that α-crystallin may exist as a biogel in the lens of the eye because the authors were able to make a gel-like substance when concentrating the protein is not very sound. This gel-like substance can be observed when concentrating any biomolecule (proteins, aptamers etc.) to extreme concentrations. Additionally, the x-ray diffraction data of the gel is reminiscent of amyloid fibril diffraction data, the authors do not present any information of how α-crystallin in fibril form would contribute to its function or how this information ties into the cryo-EM data. It also seems quite plausible that this could all just be an artifact resulting from concentrating the protein.
  5. The authors present no argument for why they are studying α-crystallin from different regions of the lens (cortex vs. nucleus). This should be explained.
  6. The authors offer no explanation for the bean shape observed in α-crystallin from the nucleus. Has this been seen before? Why would there be a change in morphology in this region? Is this significant?
  7. There are times throughout the paper when I have trouble discerning the points being made due to odd phrasing or poor sentence structure. This work requires a decent amount of English language editing.

Specific Comments

  1. The title needs to be re-worded to make sense.
  2. In the abstract, the x-ray diffraction data indicates the α-crystallin forms fibrils and is not indicative of a gel-like state in the lens. The overconcentrating is what formed a gel and x-ray data was not needed to see the gel-like state.
  3. Line 90, αL-crystallin should be changed to αA-crystallin.
  4. Line 387, the sentence beginning, “Currently, these data suggest that…”, the authors state that α-crystallin in the gel-like fibril state retains the chaperone function, but no data was presented to support that statement.
  5. Line 449, one of the authors conclusions is that it is impossible to obtain a reliable model of α-crystallin. This is a strange conclusion to present. Just because this is a very difficult protein to work with does not mean a reliable model will not be generated in the near future. 
  6. Line 453, another conclusion is that everything may be different in vivo than in vitro and further analysis is needed. This could be true of almost any biological system that is studied.  
  7. Line 230 is titled section 2.3 and should be changed to 2.4.

Reviewer 4 Report

The article “  A new view at the accepted idea of the work of α- crystalline. Own data and analysis of literary sources” by Olga M. Selivanova and Oxana V. Galzitskaya is an interesting study on alpha-crystallin, a major component of the human eye lens. They attempted to understand the detail structural aspect of this protein which is limited due to its heterogeneous nature. Several important issues need to be addressed to get clear information. My specific suggestions and comments are listed below.

  1. Abstract: lines 26-27, I did not see any experimental evidence in this paper to conclude that αA-crystallin plays the role of chaperone for αB-crystallin.
  2. Introduction:
    1. This section is too long and contains lots of redundancy that need to fix carefully. Some of the sentences are too long to keep track of information and their correlation.
    2. Line 111-116, it appeared incomplete and does not make any sense with the sentence “ taking into account…etc.
  3. Result:
    1. Supplementary fig S1 lack footnote and proper labeling. It is not mentioned what is mKg. I am not sure about the lower third band on S1A. is it a degradation or simply some impurities in preparation?
    2. Figure S1B, in-text it was mentioned that the molecular weight of αA and αB is 19.8 and 20 kDa respectively, however, gel image showing faster migration for αB protein which is wrong. Heavier protein tends to migrate slower in the gel. I am also skeptical about the resolution of two bands with 0.2kDa differences. The author should mention the percentage of the gel used.
    3. As mentioned in the method that EM samples were prepared at either 4⁰C or 37⁰C but some of the figures did not mention temperature used. Were there any differences in the ability of alpha-crystallin to form aggregate or oligomer at two different temperatures?
    4. It is not clear whether the concentration of protein has any effect on oligomerization. Does a change in the concentration of protein affect the presence of high order oligomer?
    5. Page 4, figure 1, a control missing. I would suggest to include αB protein as a control to show and compare the homogeneity between αA and αB. An absence of control making the analysis difficult.
    6. Page 5, Figure 2, includes concentration and conditions information briefly in a footnote. What was the temperature for sample preparation?
    7. Line 166-170, redundant information.
    8. Figure 3: This is an interesting image when compared to the figure 2. But at the same times open up a lot of questions, the author should analyze the figure 2 and 3 more intensely to get the proportion of the population having a different shape and size. Does this change in shape relate to different environmental conditions of cortex vs nucleus?
    9. 3D Reconstruction: some lines are redundant and line 185-193 does not fit into subject heading. This section needs to be rewritten and focus on 3D reconstruction related information. Some part in this section is very poorly written which does not make any sense like line 198-200.
    10. X-ray analysis: This section was well done with appropriate control but again missing data interpretation. Although the author talked about the diffraction pattern of X-ray but did not explain how does this pattern related to amyloid structure? Figure 5.1 should be 5. A and so on.
    11. Figure 5, fix 4,5Å to 4.5Å, same for others as well.
    12. Line 255 reference missing.
    13. Figure 5: include any control for direct comparison like Bgl2 protein.
  1. Discussion: this section has lots of redundancy with information already in the introduction section.
    1. Line 284: I did not see any data for molecular mass determination in this study and their subsequent comparison with published one however this line claim for the same.
    2. One of the biggest issues I see with this paper is the novelty of the work. Although the author obtained the model at low resolution even though they fail to produce any new structural details from data. The pieces of information mentioned in this paper are already known in this subject area.
    3. Line 308- 310, 344-345: not clear.
    4. Line 372, reference missing.
    5. Line 388, In the absence of substantial evidence, it not correct to conclude that α-crystallin present in form of amyloid in eye lens cells.
  1. Method section:
    1. Check Speed of centrifugation, 16.5/16 rpm is too low to separate.
    2. Line 425-426, not well written.
  2. Supplementary Fig S2: not footnote, no description about color and pattern. It is not clear what A and B are.

Round 2

Reviewer 2 Report

The authors have addressed my major concerns/criticisms and I recommend publication of the manuscript in Biology.

Author Response

Thanks to the reviewer. We have added new Figure 7 to the article, two additional Figures to the supplementary and the discussion about using AFM.

Reviewer 3 Report

Broad Comments

Thank you for addressing the concerns in the previous review.  I am satisfied enough with these responses. My biggest concern with this manuscript at this point is the need for significant English language editing.  While I appreciate the authors attempts to edit the previous version, many of the changes were made to sentences that were fine and now they no longer make sense.  Additionally, made of the changes simply swapped out one word for another similar word e.g. changing selected for chosen.  The main problem is in the structuring of many sentences and in many cases the words chosen are not correct for what the authors are trying to say. Below are several examples, although there are many more.

Specific Comments

  1. The second sentence of the title needs a period at the end.
  2. Line 44, keep the word “the”
  3. Line 53, keep the word “the”
  4. Line 57, your original version of this sentence was better
  5. Line 69, keep the word “the”, delete the added “a” following “the”
  6. Line 75, this sentence needs to be rewritten for English language clarity
  7. Line 92, the sentence beginning “The cryo-EM data…” keep the original version of this sentence
  8. Line 99, the sentence beginning “As shown in…”, keep the original version of this sentence
  9. Line 108, keep the original version of this sentence
  10. Line 120, this whole paragraph doesn’t make sense since you start out by saying that there is currently a model available and then end the paragraph by saying the absence of a model complicates interpretations. Please clarify.
  11. Line 128, the added section starting with “Despite intensive studies…” and ending at line 137 needs to be re-written for English language clarity.
  12. Line 255, α-crystallin is misspelled.

There are many, many additional examples like the above where the English language editing is required, particularly in the Discussion section.

Author Response

Thanks to the reviewer. We once again have corrected the text of the article.

Reviewer 4 Report

The revised version of the article “  Structural and functional peculiarities of α-2 crystalline. Own data and analysis of literary sources” by Olga M. Selivanova and Oxana V. Galzitskaya has improved but still have critical deficiencies in using controls, proper interpretation of data, languages issue and many more. Authors tried to improve the manuscript by incorporating some of the concerns mostly focused on the overall structure of the manuscript rather than working on a scientific issue related to the use of control, data interpretation, etc. I did not see much difference in a manuscript from the previous version. Most of the concerns are not addressed.

  1. The introduction and discussion section still have lots of redundancy. There is still scope to improve the discussion section. This section has not critically discussed results or previous findings.
  2. I understand that the focus of this paper was to study native a-crystallin but without control none of the data are meaningful. I strongly recommend ensuring proper control wherever it requires throughout the manuscript.
  3. Comment no 8 in the previous report, it is not addressed. Moreover, as noted in a cover letter, under point no 8, there is no reference in line no 216-217. Overall I saw a very superficial cosmetic coverup of the concern raised for the original manuscript.
  4. I apologize for the typo in my comments no 13 (in original comment) in the result section. Comment no 13 was actually for Figure 6 (instead of Figure 5). I was recommending control for this figure.
  5. Line no 359-360, 19.2S is sedimentation coefficient value, not molecular mass, there should be some number in kDa corresponding to this value.

Round 3

Reviewer 4 Report

I appreciate the author's effort to improve the manuscript and including new control data. Now the present version looks more convincing and interesting. There are minor typos in manuscripts that can be fixed during the proof-reading.